# Method Consideration of Variation Diagnosis and Design Value Calculation of Flood Sequence in Yiluo River Basin, China

**Xinxin Li [1], Xixia Ma [2], Xiaodong Li [1,3,*] and Wenjiang Zhang [1,3]**

[1]  College of Water Resource and Hydropower, Sichuan University, Chengdu 610065, China;
     xinxin1_li@163.com (X.L.); zhangwj@lreis.ac.cn (W.Z.)
[2]  School of Water Conservancy and Environment, Zhengzhou University, Zhengzhou 450001, China;
     maxx@zzu.edu.cn
[3]  State Key Laboratory of Hydraulics and Mountain River Engineering, Chengdu 610065, China
*   Correspondence: lxdscu@163.com

**Abstract:** The conventional approaches of the design flood calculation are based on the assumption that the hydrological time series is subject to the same distribution in the past, present, and future, i.e., the series should be consistent. However, the traditional methods may result in overdesign in the water conservancy project since the series has non-stationary variations due to climate change and human activities. Therefore, it is necessary to develop a new approach for frequency estimation of non-stationary time series of extreme values. This study used four kinds of mutation test methods (the linear trend correlation coefficient, Mann–Kendall test, sliding *t*-test, and Pettitt test) to identify the trend and mutation of the annual maximum flow series (1950–2006) of three hydrological stations in the Yiluo River Basin. Then we evaluated the performance of two types of design flood methods (the time series decomposition-synthesis method, the mixed distribution model) under the impacts of climate change and human activities on hydro-meteorological conditions. The results showed that (a) the design flood value obtained by the time series decomposition-synthesis method based on the series of the backward restore is larger than that obtained by the decomposition synthesis method based on the series of the forward restore; (b) when the return period is 100 years or less, the design flood value obtained by the mixed distribution model using the capacity ratio parameter estimation method is less than that obtained by the hybrid distribution model with simulated annealing parameter estimation method; and (c) both methods can overcome sequence inconsistency in design frequencies. This study provides insight into the frequency estimation of non-stationary time series of extreme values under the impacts of climate change and human activities on hydro-meteorological conditions.

**Keywords:** non-stationary; variability diagnosis; climate change; design flood

---

## 1. Introduction

Design flood is the basis for the design of wad project, whose calculation accuracy directly affects the scale and flood control safety of the project [1]. Therefore, the critical issue of design flood calculations has attracted widespread attention from the water science and hydrology communities, and the researchers have contributed a large number of formulas, methods and models [2–6].

The traditional design value calculation always uses the line fitting method based on Pearson III distribution in China, which assumes that the hydrological series are stationary over time [7,8]. However, the combination of natural weather fluctuations and extensive human intervention contributed to global climate change also significantly impacts global water resources [9–12]. The hydrological series data have become inconsistent due to the influence of climate and human activities [13].

This leads to an increase in the error of the design value obtained by the traditional hydrological frequency analysis method, and affects the design accuracy of water conservancy projects. It is not conducive to the construction and operation management of wad projects, which may cause the loss of people and property [14]. Therefore, it is of scientific significance and practical value to study the design flood calculation method of the non-stationary hydrological extreme value sequence under the changing environment.

According to the water conservancy industry standard of the People's Republic of China, "Specification for Design Flood Calculation for Water Conservancy and Hydropower Projects" (SL44-2006), if the consistency of the original flood sequence is affected by the construction of storage, diversion, and diversion project in the basin, it is necessary to check and process the consistency of the affected or mutated flood sequence and unify the hydrological data. The variability diagnosis of hydrological series detects the deterministic components in hydrological series and diagnoses whether they have the trend components, jumping components, and periodic components. The standard methods include the linear trend correlation coefficient test [15], Mann–Kendall nonparametric test [16,17], R/S analysis [18], and the Hurst index method [19], etc. Xie proposed that the periodic components are more evident on a small scale (within a year) and changeless between years, which could eliminate the influence of periodicity by selecting hydrological series with the method of annual maximum value [20]. In this paper, the annual maximum flow sequence is selected, so only trend and jump variation would be considered.

Rainfall and runoff are complex processes, so hydrologists are still struggling to find the most appropriate models and methods of frequency calculation. Several studies have shown [21–25] that hydrological calculation added geomorphological properties would make the results more accurate. Therefore, this study incorporates the evolution of land use types and changes in flows before and after construction to calibrate and validate the theoretical diagnostic results with relevant data information. A most likely point of variation is then selected for subsequent frequency calculations.

Hirsch [19] suggests that once we realize that sequence is non-stationary for various reasons, we must reconsider our planning method. There are three types of frequency analysis methods for inconsistent hydrological series in changing environments [26]: sequence reconstruction method, distribution weighted function method, and probability distribution model. The sequence reconstruction method's principle is to reconstruct the non-uniform hydrological sequence to obtain a new hydrological sequence meeting the consistency requirements, and then use the traditional hydrological frequency analysis method to calculate the design flood, which is simple and widely used [27–30]. The distribution weighted function method is to classify the hydrological series according to different periods or different background values according to the position of variation points to get consistent subsequences, then weight the distribution function of subsequences to get the frequency distribution function of the whole original series, and then calculate the design flood. This method is based on the frequency analysis function, and the result is more accurate [31,32]. The above two methods' calculation accuracy is closely related to selecting the hydrological series data's mutation points, and would be used in this paper.

This paper focuses on identifying and verifying the mutation point and evaluating two types of design flood methods.

## 2. Study Area

Yiluo River is a first-class tributary on the Yellow River's right bank, crossing Henan and Shaanxi provinces. The total length of the river is 447 km, and the drainage area is 18,881 km$^2$. The main tributaries are the Luohe River, Yihe River, and Yiluo River section (Figure 1). The whole basin's topography is high in the West and low in the East, with an average gradient of 3.7‰. Further, the arable land is nearly half of the total watershed area. The Yiluo River belongs to a warm temperate continental monsoon climate, with hot and humid summer and cold and dry winter. The average annual temperature is 12 °C. Annual precipitation is 700–900 mm, about half concentrated in July to

September. The average annual total water resource is 3.221 billion m$^3$, and the average perennial runoff flow is 883 million m$^3$, with the average perennial runoff depth change range between 250–327 mm.

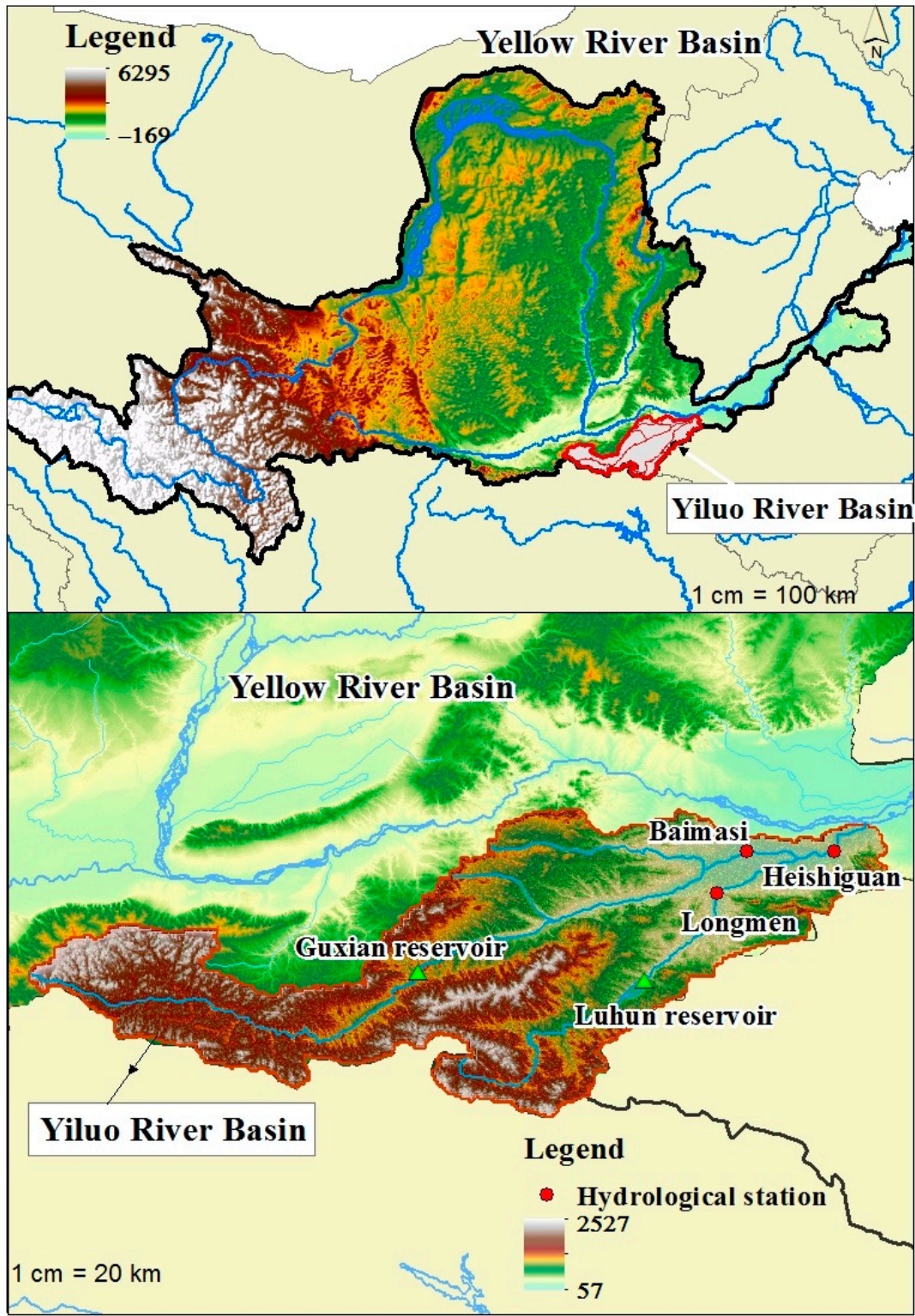

**Figure 1.** Location of study area and gauging stations.

There are many completed water conservancy projects in the Yiluo River Basin, including two large reservoirs, Luhun reservoir and Guxian reservoir. Luhun reservoir was built in 1960 and completed in

1965, controlling the drainage area of 3492 km$^2$. Guxian reservoir reservoir was started in 1985, which was shut down and resumed for many times due to various reasons. It was not built until 1993, and put into use about 1994, controlling the drainage area of 5370 km$^2$.

The Yiluo River Basin has important hydrological stations such as Baimasi, Longmen, and Heishiguan. Heishiguan stations (112°56′ E, 34°43′ N) was established in Gongyi, Henan Province in July 1934. It belongs to the Yiluo river system of the Yellow River Basin. The catchment area of the station is 18,563 km$^2$, which is an important gateway to Yiluo. Baimasi station is located in the Luohe River, with a control basin area of 11,891 km$^2$, accounting for 98.8% of the Luohe River Basin area; Longmen Town station has a control basin area of 5318 km$^2$, accounting for 88.2% of the Yihe River Basin area.

## 3. Data

### 3.1. Basin Underlying Surface Data

The 1-km grid data of land use types across three periods of 20 years (the end of 1970s, the end of 1980s, and 1995) were from the data center of resources and environmental science, Chinese Academy of Sciences (http://www.resdc.cn/). Data of land-use cover classification system was used and included six first classed types of cultivated land, forestland, grassland, water area, residential land, and unused land, and 25-s class types from China Academy of water resources and hydropower (http://www.dsac.cn/ServiceCase/Detail/413985).

### 3.2. Hydrometeorological Data

The complete maximum annual discharge data of Heishiguan station from 1951 to 2006, Baimasi station from 1952 to 2006, and Longmen station from 1950 to 2006 used in the study were provided by the hydrological Manual of Henan Province. The long-term hydrological series data was more than 30 years, highly representative. According to the investigation, the historical super flood in 1961 had maximum flow of 20,600 m$^3$/s, and the return period was more than 400 years [33].

## 4. Methodology

### 4.1. Trend Detection Method

#### 4.1.1. Linear Trend Test

The linear trend test is extensively used to determine the trend of hydrological series. The following equation expresses its trend [34]:

$$x_t = a + bt + \varepsilon_t, \quad t = 1, 2, \cdots, n \tag{1}$$

where $a$ and $b$ is the parameter constant, $\varepsilon_t$ is the residual value of $a + bt$ and the true value $x_t$, $n$ is the length of hydrological sequence. In order to get the best fitting and the least residual value, the least square method is used to estimate the parameters $a$ and $b$.

#### 4.1.2. Mann–Kendall Test

The Mann–Kendall test (MK-test) is a typical non-parametric test method [35]. It is widely used in the trend diagnosis of rainfall, temperature and other hydrological series. The principle of MK-test is as follows: assuming hydrological series $x_i, x_j$, where $i = 1, 2, \ldots, n, 1 \leq j < i$, $s_k$ is the cumulative value of $x_i > x_j$, i.e.,

$$s_i = \begin{cases} +1, & x_i > x_j, \\ 0, & x_i \leq x_j, \end{cases} \quad j = 1, 2, \cdots, i \tag{2}$$

$$s_k = \sum_{i=1}^{k} s_i, \ k = 2, 3, \cdots, n \tag{3}$$

The formulas of statistical $UF_k$ and $UB_k$ are that:

$$UF_k = \frac{s_k - E(s_k)}{\sqrt{Var(s_k)}}, \ k = 2, 3, \cdots, n \tag{4}$$

$$UB_k = -UF_k, \ k^* = n + 1 - k \tag{5}$$

where $E(s_k)$, $Var(s_k)$ is the mean value and variance of $s_k$, respectively. Given the significance level $\alpha$, two thresholds can be plotted. If two lines exceed the boundary line of two thresholds obviously, it indicates that the original sequence has obvious trend variation, and the time point corresponding to the intersection point is the abrupt change point.

### 4.2. Jumping Detection Method

#### 4.2.1. Hurst Index Method

The Hurst index method based on R/S (Rescaled range analysis) is to divide the hydrological series into $M$ short series with the length of $N$, and the sum of the subsequences' length is the length of the original series, $n = M \times N$. Mandelbrot has proved through many experiments that there is a certain relationship between the standard deviation $S$ and $R$:

$$R/S = K(n)^{Hurst} \tag{6}$$

where, $R = \max(x_{i,k}) - \min(x_{i,k})$, $K$ is a constant, and *Hurst* coefficient is unknown. Take the logarithm of Equation (6) to get:

$$\ln (R/S)_n = Hurst(\ln n) + \ln(K) \tag{7}$$

Generally, it is considered that *Hurst* = 0.5 indicates that the time series is random. Otherwise, it indicates that has jumping or other variations. The farther it is from 0.5, the greater the variation of sequence.

#### 4.2.2. Sliding T Test

The sequence $x_i = \{x_1, x_2, \cdots, x_n\}$ is divided into two subsequences, $x_i = \{x_1, x_2, \cdots, x_{n_1}\}$ and $x_j = \{x_1, x_2, \cdots, x_{n_2}\}$, where $n_1 + n_2 = n$. The mean value formula of the subsequence is as follows:

$$\bar{x}_1 = \frac{1}{n_1} \sum_{i=1}^{n_1} x_i \tag{8}$$

$$\bar{x}_2 = \frac{1}{n_1} \sum_{j=1}^{n_2} x_j \tag{9}$$

According to statistics [36], the statistics $T$ of this sequence is

$$T = \frac{(\bar{x}_1 + \bar{x}_2)}{(\frac{1}{n_1} + \frac{1}{n_2})[(n_1 S_1 + n_2 S_2)/v]^{\frac{1}{2}}} \tag{10}$$

Sliding T test (*t*-test) needs to test the segmentation points one by one according to the time sequence. The segmentation point corresponding to T with the largest absolute value is the most likely variation point.

### 4.2.3. Pettitt Test

Pettitt test is a nonparametric variation point test proposed by Pettitt in 1979. It can test the significance of variation and the position of variation point simultaneously [37]. The formula of statistic $U_{t,n}$ is as follows:

$$U_{t,n} = U_{t-1,n} + S_{t,n}, \quad t = 2, 3, \cdots, n \tag{11}$$

where,

$$S_{t,n} = \sum_{j=1}^{n} \text{sgn}(x_t - x_j) \tag{12}$$

$$\text{sgn}(x_j - x_i) = \begin{cases} +1, x_j > x_i \\ 0, x_j = x_i \\ -1, x_j < x_i \end{cases} \tag{13}$$

The position corresponding to the maximum value of $K_\tau$ is the possible jumping point,

$$K_\tau = \max_{1 \le t \le n}(|U_{t,n}|) \tag{14}$$

The significance level of variation was determined by the following formula,

$$p \cong 2 \exp\left[-6(K_\tau)^2 / (n^2 + n^3)\right] \tag{15}$$

when $p < 0.05$, it indicates that the location is the jumping point, which is the first-order variation point of the sequence.

### 4.3. Frequency Analysis

#### 4.3.1. Time Series Decomposition-Synthesis Method

Xie Ping [14] proposed the basic hypothesis of time series decomposition-synthesis method in 2005, which consists of two parts: random component and deterministic component.

$$Q_x = \begin{cases} S_x, & 1 \le x < x_0 \\ S_x + Y_x, & x_0 \le x \le n \end{cases} \tag{16}$$

where $S$ is the random component, $Y$ is the deterministic component, $Q_x$ is the measured time series, $x$ is the year, and $x_0$ is the year of the variation point.

The time series decomposition synthesis method includes forward restore and backward restore. Forward restore (decomposition) means the series before the mutation point is corrected to be consistent with the series after the mutation point. Backward restore (synthesis) means the series after the mutation point is corrected to be consistent with the series before the mutation point.

#### 4.3.2. Mixed Distribution Model

The mixed distribution model can be expressed as:

$$F(x) = \sum_{i=1}^{n} \alpha_i F_i(x) = \alpha_1 F_1(x) + \alpha_2 F_2(x) + \cdots + \alpha_n F_n(x) \tag{17}$$

where, $F_i(x) = \{F_1(x), F_2(x), \cdots, F_n(x)\}$ is cumulative distribution function of $n$ subsequences, $\alpha_i = \{\alpha_1, \alpha_2, \cdots, \alpha_n\}$ is the weight of the subsequence, and $1 = \sum_{i=1}^{n} \alpha_i = \alpha_1 + \alpha_2 + \cdots + \alpha_n$.

Alila Y. and Mtiraoui A. [31] pointed out that two aspects should be paid attention to ensure model parameter estimation accuracy when dividing the subseries. On the one hand, to ensure that the

subseries obeys the corresponding distribution function, on the other hand, it is necessary to strictly control the number of subseries to maintain a minimum, to improve the accuracy of the model. In this study, the sequence is split into front and back subsequences according to the most likely variation point. Assuming the probability distributions of subsequences satisfy the Pearson III distribution.

$$f(x) = k f_1(x) + (1-k) f_2(x) \tag{18}$$

$$f_i(x) = \frac{\beta_i^{\alpha_i}}{\Gamma(\alpha_i)} (x - a_{0i})^{\alpha_i - 1} e^{-\beta_1 (x - \alpha_{0i})} \tag{19}$$

where $k$ is the weight of the subsequence; $\alpha_i, \beta_i, a_{0i}$ is the parameter of the Pearson III probability density function $f(x)$. $i = \{1, 2\}$, and $\Gamma(\alpha_i)$ is the gamma function.

At present, many scholars at home and abroad have studied the parameter estimation of the hybrid distribution model, such as nonlinear optimization, simulated annealing algorithm (SAA) [38] and the maximum likelihood EM algorithm (Expectation-Maximization algorithm, EM). Simulated annealing is a nonlinear method based on Monte Carlo iteration, which is suitable for global optimization [39]. Therefore, this paper uses SAA to estimate the parameters.

## 5. Results and Discussion

### 5.1. Variation Diagnosis

#### 5.1.1. Trend Diagnosis

When the significance level is 0.05, the trend lines of annual maximum flows sequence of Heishiguan, Baimasi and Longmen stations show downward trend with obvious trend variation (Figure 2a,c,e). It was further verified by the Mann–Kendall test. Figure 2b,d,f all show that the UF began to exceed the threshold line in 1969 and showed a continuous decline.

#### 5.1.2. Jump Diagnosis

The Hurst index with significant level 0.05 is shown in Table 1. Overall, variation occurred at all three sites in the basin, while Heishiguan and Baimasi station with more strong mutation.

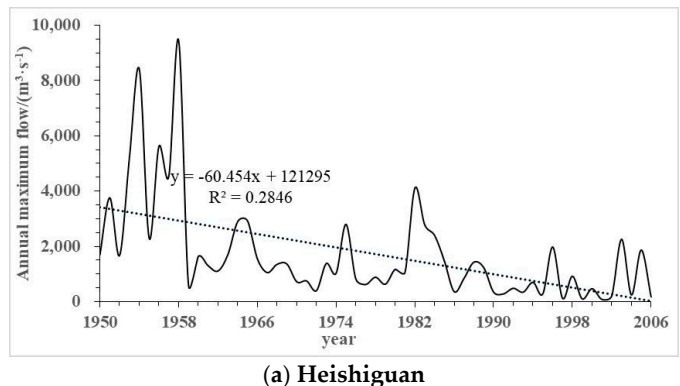 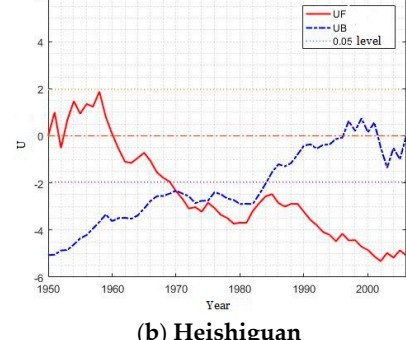

(a) Heishiguan　　　　　　　　　　　　　　　　　(b) Heishiguan

**Figure 2.** *Cont.*

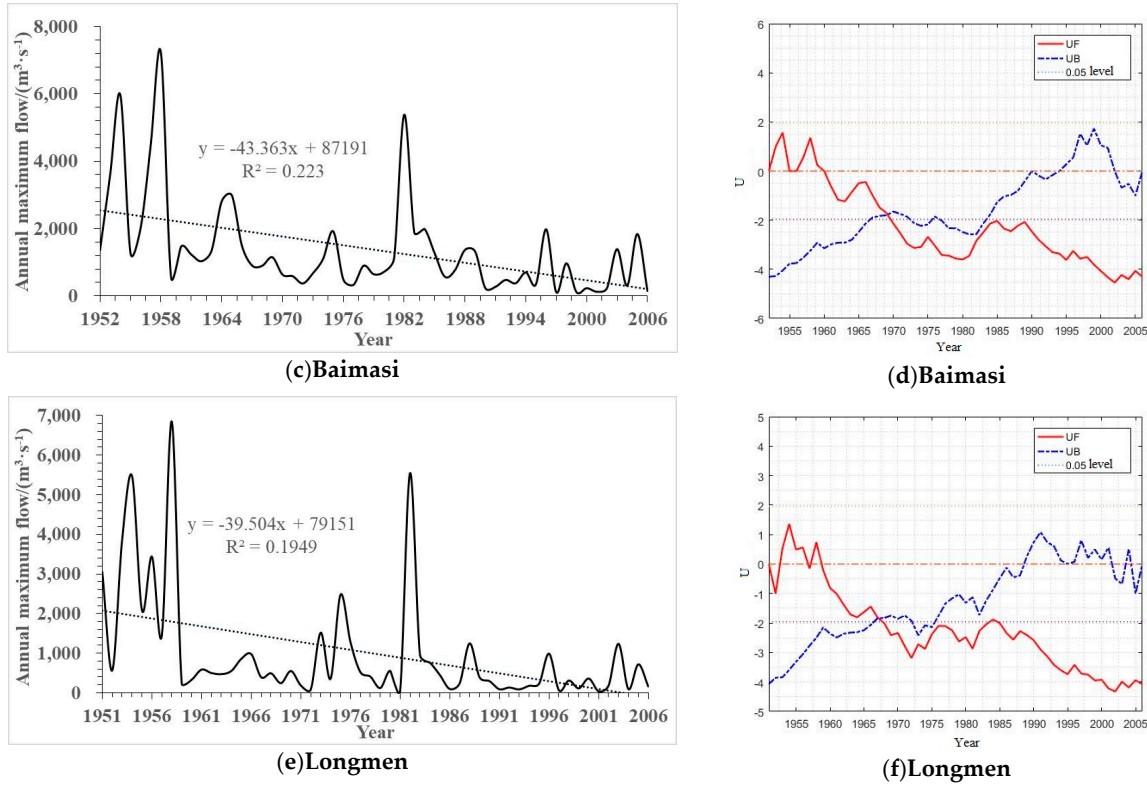

**Figure 2.** Trend of annual maximum discharge sequence in Heishiguan, Baimasi and Longmen hydrological station ((**a**,**c**,**e**) trend test, (**b**,**d**,**f**) Mann–Kendall test).

**Table 1.** Hurst index and variability.

| Station | Hurst Index | Variability |
|---|---|---|
| Heishiguan | 0.8905 | strong |
| Baimasi | 0.8778 | strong |
| Longmen | 0.7694 | moderate |

The largest statistics T of the sliding *t*-test is in 1969, 1989 and 1985, Figure 3a, in Heishiguan station. The Pettitt test was used to further diagnose the sequence. In Figure 3b, the peak value exceeds the threshold value, indicating that the Heishiguan station's sequence has a jumping mutation, and the possible variation point is in 1960–1992. In the same way, the strongest change of Baimasi and Longmen station occurred in 1989 and 1966 respectively from Figure 3c,e that sliding *t*-test. Judgments on the Pettitt test for Figure 3d,f exceeded the threshold for 1964–1976 and 1982–1991, 1975–1980 and 1982–1990 respectively, with the most likely point of variation in 1989 in Baimasi and Longmen station.

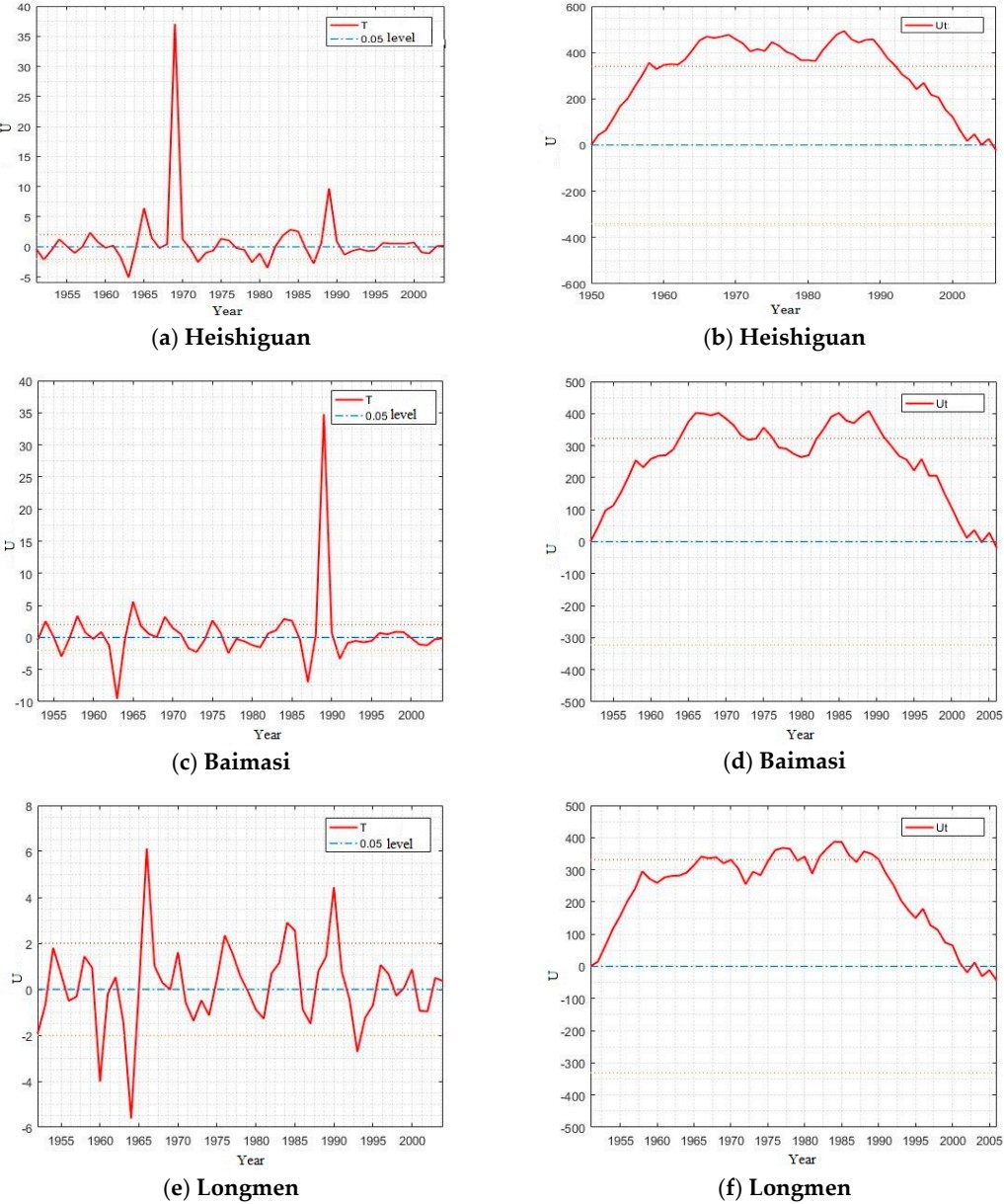

**Figure 3.** Jumping of annual maximum discharge sequence in Heishiguan, Baimasi and Longmen ((**a**,**c**,**e**) are the sliding *t*-tests; (**b**,**d**,**f**) are Pettitt tests).

### 5.1.3. Combination of Variation Diagnosis

The results of the above variation diagnostic methods (Table 2) are theoretically possible variation points. The most likely variation point obtained by combining the three sites and multiple methods is in 1989. The time of the variation point needs to be verified and tested against physical factors, such as constructing large hydraulic projects and the evolution of land use types, to ensure accuracy for the subsequent frequency analysis.

**Table 2.** Variation points and degree.

| Diagnostic Type | Diagnostic Method | Diagnostic Results | | |
| --- | --- | --- | --- | --- |
| | | **Heishiguan** | **Baimasi** | **Longmen** |
| Trend diagnosis | Linear trend test | Remarkable (↓[1]) | Remarkable (↓[1]) | Remarkable (↓[1]) |
| | Mann–Kendall test | Remarkable (↓[1]) 1969 | Remarkable (↓[1]) 1969 | Remarkable (↓[1]) 1969 |
| Jump diagnosis | Hurst index method | Strong variation | Strong variation | Moderate variation |
| | Sliding *t*-test | Remarkable 1969, 1989 *,[2], 1985 | Remarkable 1989 *,[2], 1963, 1987 | Remarkable 1966, 1964, 1984 |
| | Pettitt test | 1960–1992 | 1964–1976, 1982–1991 | 1975–1980, 1982–1990 |

[1]↓ indicates a downward trend; [2,]* is the most likely variation point.

The literature [40] shows that Luhun reservoir was built in 1965 in Yiluo River Basin, which control and guidance projects were added in 1969. The controlled area of the Luhun reservoir is 3492 km$^2$. The Guxian reservoir was first started built in 1958, but it was shut down and resumed many times. From 1978 to 1993, many temporary works such as diversion and trestle were built in the reservoir. The control area of the Guxian reservoir is 5370 km$^2$, and the total capacity is 1.1758 billion m$^3$. It was considered that after the construction of a large-scale water conservancy project, its operation would affect the flow. For example, at Heishiguan station, taking 1965 and 1993 as nodes, the annual maximum flow sequence is divided into three sections (Figure 4a), 1950–1965, 1965–1993, and 1993–2006. It can be seen that the construction of water conservancy projects will reduce the maximum discharge to a greater extent. Because of the early operation and smaller control area in the Luhun Reservoir basin, the reservoir's regulatory capacity may have been curtailed under the 57-year long sequence in the study. The sequence's inconsistency after the construction of the Luhun reservoir (post-1965) can be considered to be generated primarily by the Guxian reservoir. Then, the change of sequences from Heishiguan station after 1965, 1965–1992 (b), 1993–2006 (c), 1965–1989 (d), were studied in Figure 4. The hydrological sequence in 1965–1992 (Figure 4b) is decreasing, but the trend line is increasing in Figure 4c, which represents that the Guxian reservoir regulated the sequence in 1993–2006. The trend line of the sequence from 1965 to 1989 tends to horizontal, in Figure 4d, so 1989 is the most likely mutation point probably. It is consistent with the theoretical diagnosis and speculated that one reason for the variance is the watershed's alteration during the Guxian Reservoir construction.

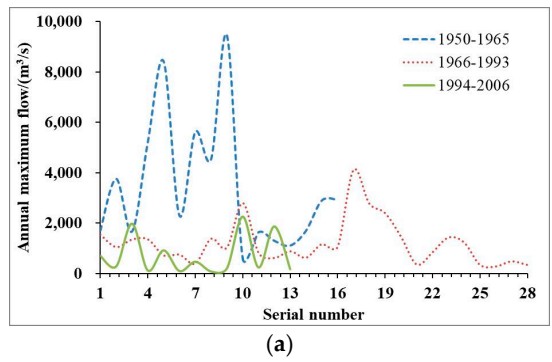

(a)

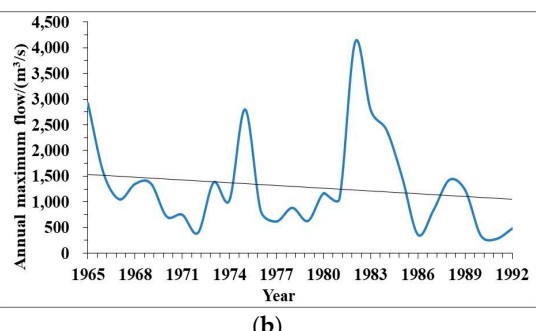

(b)

**Figure 4.** *Cont.*

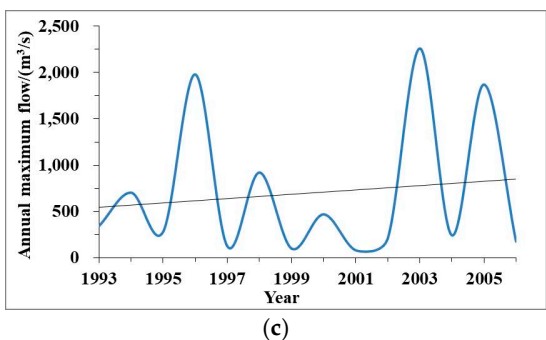 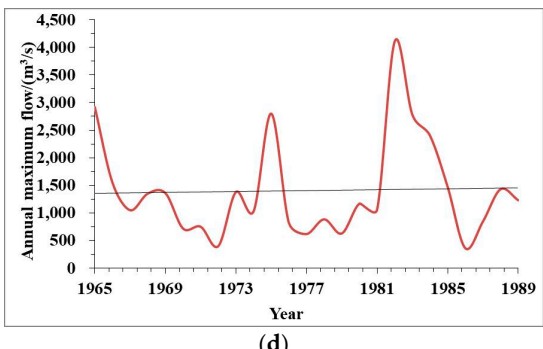

(c)             (d)

**Figure 4.** Sequence diagram of annual maximum flow. (**a**) According to the construction time of the project; (**b–d**) according to the theoretical variation point.

The main driving forces of rivers are precipitation and underlying surface conditions of basins, so the change of flow series is caused by natural climate change and human activity change [41]. In the 1980s, human activities and climate change reduced runoff in the Yiluo River Basin. The main factor in climate change is the decrease in rainfall, which impacted runoff by 58% [42].

Land use factors include modern urbanization, forest, agriculture, and animal husbandry management. These factors can affect hydrological series for several centuries in time and greatly influence spatial scale [43,44]. Land cover change is considered the cause of flood variation point or trend variation [45]. Increases in building masses and waters can significantly rise in evaporation [46,47]. Residential areas form the impervious surface area. Hammer [48] demonstrated that when the impervious area reaches 10% of the watershed, it will significantly impact the hydrological processes. Olivera [49] showed that when the impervious rate is 10%, the annual runoff and peak flood flow increases by half. Brown [50] found that land use and land cover will alter watershed runoff, with the impervious area being proportional to runoff.

Basin variability is estimated by investigating the rate and timing of change in surface cover, supported by literature findings on land use and watershed change response [43–50]. The terrain of Yiluo River basin which is narrow and long is high in the West and low in the East. The western and southern shores are densely forested with many vegetation types. The eastern side is flat and multi-building, located in the lower reaches of the river. There are heavy rains in summer. Most of the valley belt is cultivated land with poor surface coverage and serious soil erosion. Land use grid data for the Yiluo River Basin in the late 1970s (a), the late 1980s (b), and 1995 (c) have been shown in Figure 5. Compared with the end of the 1970s, the area of water area and residential building area in the basin is increasing, and the area of residential area is increasing the fastest. Meanwhile, the area of forest land and grassland is decreasing, and the rate of grassland reduction is the fastest. The decrease in forest and grassland cover will reduce the retention of rainfall and increase runoff. It can be seen that the underlying surface of the basin is constantly changing, leading to changes in runoff and evaporation within the watershed. According to Table 3, the rate of change is more pronounced after the 1980s. The period in which land use change is presumed to have caused significant changes in the hydrologic series is the late 1980s. This is consistent with the results of the theoretical diagnostic approach. However, natural variations in global climate have not been studied. The uncontrolled changes in the rivers' characteristics and non-removable errors due to data resolution are among the uncertainties in runoff simulations. Since the land cover type is used only as a test and calibration of the theoretical variability time, this does not involve any research methodology parameters in this paper. The presence of uncertainties is considered to not have a significant impact on the validation process.

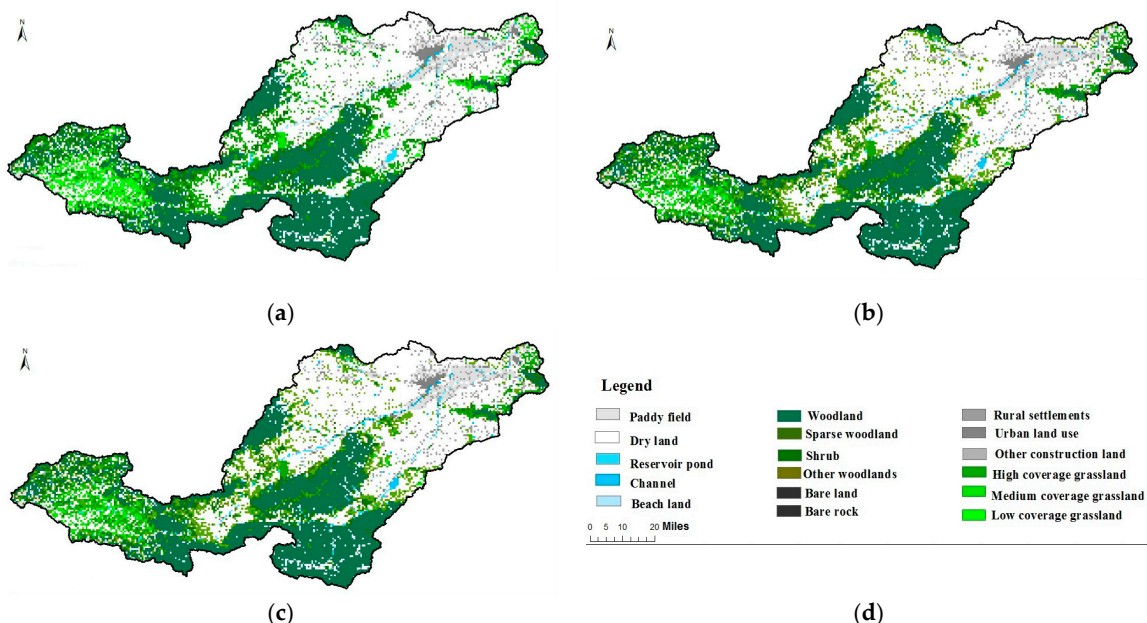

**Figure 5.** Spatial distribution of land use types in Yiluo River Basin in different periods. (**a**) In late 1997; (**b**) in the late 1980s; (**c**) in 1995; (**d**) legend.

**Table 3.** Proportion and relative change rate with the end of 1970s of land use types in Yiluo River Basin in different periods.

| Land Use Type | Area Ratio/% | In the Late 1970s | In the Late 1980s | Change Rate | In 1995 | Change Rate |
|---|---|---|---|---|---|---|
| Cultivated land | Paddy field<br>Dry land | 44.17 | 44.13 | −0.11 | 43.95 | −0.5 |
| Woodland | Woodland<br>Shrub<br>Sparse woodland<br>Other woodlands | 34.14 | 34.1 | −0.11 | 33.96 | −0.53 |
| Grassland | High coverage grassland<br>Medium coverage grassland<br>Low coverage grassland | 16.43 | 16.42 | −0.1 | 16.15 | −1.74 |
| Waters | Channel<br>Reservoir pond<br>Beach land | 1.57 | 1.61 | 2.36 | 1.7 | 8.08 * |
| Residential area | Urban land use<br>Rural settlements<br>Other construction land | 3.65 | 3.71 | 1.74 | 4.21 | 15.38 * |
| Unused land | Bare land<br>Bare rock | 0.03 | 0.03 | 0 | 0.03 | 0 |

\* mean major changes.

Physical analysis from two aspects of the changing environment (climate change and underlying surface change) verifies and tests the variability diagnosis results. It is determined that the period of evident variation of hydrological series caused by the changing environment is the end of 1980s, which is consistent with the results of theoretical diagnosis method.

*5.2. Design Value*

5.2.1. Time Series Decomposition Synthesis

The annual maximum flow hydrograph after trend decomposition is gentler than before (Figure 6a). When $\alpha = 0.05$, the diagnosis results showed that there was no significant variation in the decomposed sequence. After the jump decomposition, the process line of the subsequence from 1989 to 2006 becomes gentler, and Figure 6b shows that the linear trend approaches the level. The significance level was 0.05, and the diagnosis of variability indicated that there was no remarkable variation in the sequence after jumping decomposition. Therefore, the time series decomposition method is feasible for this series. The design flood calculation can be carried out based on the decomposed series.

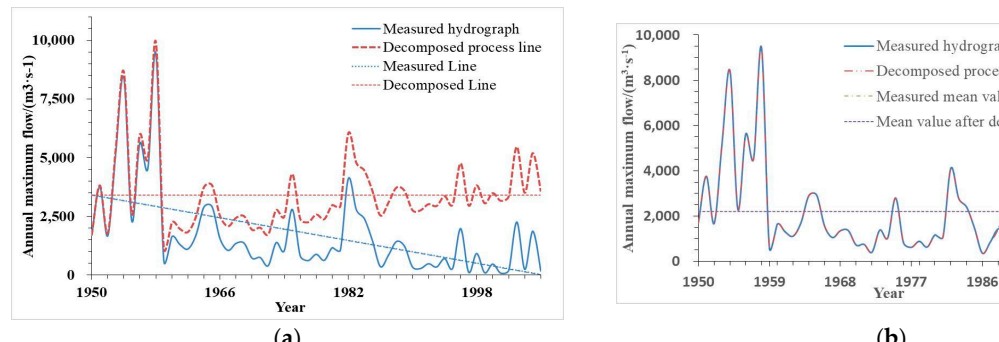

**Figure 6.** Time series before and after decomposition. (**a**) Trend, (**b**) jump.

The year 2006 was used as the base year, and the trend and jump synthesis of annual maximum flow series were carried out. Using Monte Carlo, 30,000 samples of Pearson III distribution satisfying trend decomposition state and jump decomposition state were randomly generated. Following this, the generated random sequence and the deterministic component together formed the random sequence under the current year's condition. Because the new sequence meets the consistency requirements, the design floods value was calculated by the traditional line fitting method.

5.2.2. Mixed Distribution Model

The capacity ratio method uses the weight of subsequence in the whole sample sequence as the distribution function coefficient. The mixed distribution equation of the original sequence based on the capacity ratio method is as follows:

$$F(x) = \frac{39}{57}F_1(x) + \frac{18}{57}F_2(x) = \frac{13}{19}F_1(x) + \frac{6}{19}F_2(x) \tag{20}$$

Referring to the parameters of Yi River and Luo River, the boundary constraint conditions are determined, and the range of $C_s/C_v$ is 2 to 3. Taking the minimum square sum of frequency dispersion as the objective function, the parameter estimation results of SAA are shown in Table 4.

**Table 4.** Parameters of the simulated annealing algorithm (SAA).

| Subsequence | Weight | Mean Value | $C_v$ | $C_s/C_v$ |
|---|---|---|---|---|
| 1950–1988 | 0.83 | 2554 | 1.22 | 2.90 |
| 1989–2006 | 0.17 | 988 | 0.91 | 2.43 |

5.2.3. Design Value Analysis

According to Table 5, the reduction state (decomposition) is larger than that of the return state (synthesis) no matter in trend state or jump state. When the return period is greater than or equal to

50 years, the flood's design value obtained by the time series synthesis method is smaller than that of the line fitting method (the error is negative). However, when the return period is less than 50 years, except for the jump decomposition method, the flood's design value obtained by the line fitting method is more extensive (the error is positive). Generally speaking, the time series decomposition synthesis method focuses more on flood design and protection in a small recurrence period, suitable for medium and small projects, especially those with a recurrence period of fewer than 50 years. The design errors of the trend decomposition and synthesis calculation of the sequence are larger than the jumping range, which shows that the sequence's trend variation is more substantial than the jumping variation. Therefore, the possibility of trend variation in the future is greater. Thus, it needs to be considered in the engineering design.

**Table 5.** Design value comparison between traditional method and other methods.

| Recurrence Period/(Year) | | 500 | 200 | 100 | 50 | 10 |
|---|---|---|---|---|---|---|
| Qm /(m$^3$·s$^{-1}$) | | 20,441.39 | 16,637.47 | 13,825.08 | 11,087.77 | 5201.4 |
| Traditional Line Fitting Method | | | | | | |
| Time Series Decomposition and Synthesis | Trend decomposition | 17,671.21 | 14,964.56 | 12,945.39 | 10,958.57 | 6545.42 |
| | E/% | −15.68 | −11.18 | −6.8 | −1.18 | 20.53 |
| | Trend synthesis | 17,074.03 | 14,102.8 | 11,892.52 | 9725.01 | 4957.7 |
| | E/% | −19.72 | −17.97 | −16.25 | −14.01 | −4.92 |
| | Jump decomposition | 18,958.87 | 15,632.26 | 13,165.37 | 10,755.45 | 5514.66 |
| | E/% | −7.82 | −6.43 | −5.01 | −3.09 | 5.68 |
| | Jump synthesis | 18,063.78 | 15,097.35 | 12,873.16 | 10,671.52 | 5698.09 |
| | E/% | −13.16 | −10.2 | −7.39 | −3.9 | 8.72 |
| Mixture Distribution Model | Capacity ratio | 19,369.16 | 15,735.77 | 13,054.66 | 10,451.43 | 4895.59 |
| | E/% | −5.53 | −5.73 | −5.9 | −6.09 | −6.25 |
| | SAA | 20,392.7 | 16,607.53 | 13,813.48 | 11,099.5 | 5300 |
| | E/% | −0.24 | −0.18 | −0.08 | 0.11 | 1.86 |

The estimation of the parameters will affect the results of the mixed distribution model. There is a difference between the flood values after the sample size ratio method and simulated annealing estimation parameters (Table 5). It should be noted that the difference between the design value obtained by the capacity ratio method and the line fitting method is larger than that by the simulated annealing method and the line fitting method. When the return period is less than 100 years, the hybrid distribution model based on simulated annealing may increase the engineering cost in the sense of engineering safety. However, it improves the protection ability of small and medium-sized projects, reduces the risk of a dam break, and is more suitable for designing small and medium-sized projects.

In conclusion, both methods can overcome sequence inconsistency in design frequencies. However, the results of the time series decomposition synthesis method are too large in amplitude with the results under smooth conditions (or with the traditional method), indicating that is not stable and needs to be explored further. The design values of the mixed distribution model (including capacity ratio and simulated annealing) under multiple recurrence intervals have errors within 10% of those obtained by the traditional method, indicating that they are stable and can meet the frequency requirements for non-consistent sequences under changing environments. From the comparative study of the above design values, it is impossible to determine which method is more effective than other methods [51]. Each method has its applicable conditions and scope. The uncertainties in each method also need to be considered.

E in the Table 5 refers to the relative error percentage of the design value of the corresponding method and the traditional line fitting method.

$$E = 100\frac{Q_{mi} - Q_m}{Q_{mi}} \tag{21}$$

where $Q_{mi}$ is the design value of time series decomposition or synthesis and mixture distribution model, and $Q_m$ is the design value of traditional line fitting method.

## 6. Conclusions

This paper aims to test whether there is a variability of hydrological series under the environmental changes, and to evaluate the performance of two types of design flood methods. For this purpose, we diagnosed the variability by several methods, including the linear trend test, the Mann–Kendall test, Hurst index, sliding *t*-test, and the Pettitt test, and compared the design values of the annual maximum flow series of time series decomposition synthesis and mixture distribution model in the Yiluo River basin. The results show that the series of Yiluo River Basin had different degrees of decreasing trend and jump variation in Heishiguan, Baimasi and Longmen station. The most likely mutation point was determined and verified is in 1989. Besides, we have solved the problem of sequence variation and inconsistency in frequency analysis by using the time series decomposition-synthesis and the mixture distribution model. However, the results of the time series decomposition synthesis method are too large in amplitude with the results under smooth conditions (or with the traditional method), indicating that is not stable and needs to be explored further. According to the calculations in this study, the hybrid distribution based on the simulated annealing method is more suitable for the design with a return period of fewer than 100 years. It could improve the protection ability of small projects and reduce the risk of dams broke.

The novelty is inferring the approximate period of river variability by studying the effects of land use type change and engineering on watersheds, supporting the calibration and validation of theoretical results. We provide an idea of the still unclear response mechanisms between natural changes and watersheds. The study raises a warning for current studies based on hydrological sequences, which require attention and consideration of their variability. Simultaneously, the study provides ideas for frequency estimation of non-stationary time series extremes under the influence of climate change and human activities on hydro-meteorological conditions. The design value is calculated on the premise that the return period is assumed to be stable [52], while the literature shows that the recurrence interval is modified or replaced by design life levels of the project [53]. It is the limitation of this paper, which needs to be discussed and studied in detail. Moreover, we also need to consider the uncertainties of each method.

**Author Contributions:** Conceptualization, X.M. and X.L. (Xinxin Li); methodology, X.L. (Xinxin Li); software, X.L. (Xinxin Li); validation, X.L. (Xinxin Li), X.M. and X.L. (Xiaodong Li); formal analysis, X.L. (Xinxin Li); investigation, X.M.; resources, X.M.; data curation, X.L. (Xinxin Li); writing—original draft preparation, X.L. (Xinxin Li); writing—review and editing, X.L. (Xinxin Li) and X.L. (Xiaodong Li); visualization, X.L. (Xinxin Li); supervision, W.Z.; project administration, X.L. (Xinxin Li); funding acquisition, X.L. (Xinxin Li). All authors have read and agreed to the published version of the manuscript.

**Funding:** This work is supported by the Key R&D projects of the Science and Technology department in Sichuan Province grant 2018SZ0343 & 2020YFQ0013.

**Conflicts of Interest:** The authors declare no conflict of interest.

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
