# Peer review of "Method Consideration of Variation Diagnosis and Design Value Calculation of Flood Sequence in Yiluo River Basin, China"

_water, doi:10.3390/w12102722_

Round 1
Reviewer 1 Report
A BRIEF SUMMARY
I suggest that the authors try to add some more references especially in the "part 1 (introduction)" of the paper. I have indicated some suggestions for specific sections of paper, but more can be added which would make the foundation for the arguments stronger.
In the meantime, I think that the authors should take into account some specific comments that follow, which, to my opinion, will improve the work.
SPECIFIC COMMENTS
Paragraph 1 (Introduction). It is interesting to consider following paper on the hydrograph estimation in ungauged watershed:
- Grimaldi, S., Petroselli, A., Serinaldi, F. 2012a. A continuous simulation model for design-hydrograph estimation in small and ungauged watersheds [Un modèle de simulation continu pour l'estimation d'hydrogrammes de projet sur des petits bassins versants non jaugés] Hydrological Sciences Journal, 57 (6), 1035-1051.
- Grimaldi, S., Petroselli, A., Serinaldi, F. 2012b. Design hydrograph estimation in small and ungauged watersheds: Continuous simulation method versus event-based approach. Hydrological Processes, 26 (20), 3124-3134.
- Petroselli, A., Grimaldi, S. 2018. Design hydrograph estimation in small and fully ungauged basins: a preliminary assessment of the EBA4SUB framework. Journal of Flood Risk Management, 11, S197-S210.
- Petroselli A., Asgharinia S., Sabzevari T., Saghafian B. 2020a. Comparison of design hydrograph estimation methods for ungauged basins in Iran. Hydrological Sciences Journal, 65(1), 127-137.
Fig. 1: I suggest to add a window with a “large scale positioning” of study area.
Fig. 2-3-4-6: the axis values are not clear. I suggest increasing the font size.
Paragraph 6 (conclusion). I suggest to rename the paragraph as “Conclusions”. I suggest also to improve this paragraph with novelty of presented work.
Reviewer 2 Report
The paper is addressing a complex issue related to definition of design hydrological values for catchment that are affected by major land uses changes and climate evolution. The paper presents a series of methods that could be used to defined design values following implementation of major reservoirs that have strongly affected the river regime of the Yiluo river.
If the objectives of the paper are clear, the current presentation has to be strongly improved in order to match the publication standards.
The presentation of the Yiluo catchment is insufficient. Data about precipitations, temperatures, evapotranspiration should be represented in details in order to understand the hydrological situation. At the same time, the implementation of the major reservoirs and their operating rules should be presented in details. Currently the impact of infratstructures looks strongly dominating and the impact of climate change is not demonstrated in a convincing way. In a similar way, the land uses evolution are not properly addressed as the evolution of each category at teh catchment scale is nor presented. The identified categories should be presented and the associated uncertainty in the identification process as well. The resolution used plays a major role and may generate significant error. This point should be addressed.
Quality of hydrological data and especially liminimetric data is not discussed. The data from a single station are presented. It would be useful to have additional elements on the other stations and to compare records from various stations in order to identify the trend.
Statement about super flood (line 115) should be clarified. This flood event should be presented with frequency and return period. At the same, the hydrological time series should be presented and quality should be assessed properly.
The conclusion should be completely revised and should make clear the added value often approach and the main results that are currently highly questionnable.
An extensive editing of the paper is needed in order to polish style and content. Help from a native speaker would be very helpful.
Reviewer 3 Report
This was an enjoyable paper to read. Thank you.
I do not know if it was a formatting issue or not, but it would be nice to clean up some of the mathematical notation and when numbers are in subscripts or superscripts (think of the times you mention square kilometers or cubic kilometers per second).
Do you have a citation for handling the mixed distribution in the manner that you do? The reason I ask is that I am so used to seeing mixture models be a combination of weighted probability density functions instead of the cumulative distribution functions that you present.
I feel the results in Table 4 could be presented better than they are. You do mention the design errors but it might be nice for the reader to see how %E is calculated for Table 4. Also there might be a cleaner way of writing Qm for each of the techniques instead of writing Qm1, Qm2, Qm3, Qm4, Qm5, and Qm6. Again, that makes things a little difficult to read.
These estimates of the design value composition are just that, point estimates. You state in Line 299 "[T]he uncertainties in each method also need to be considered." Is this future work for the authors?
Round 2
Reviewer 2 Report
Most of the comments have been integrated within the new version of the paper. However some aspects within the methodology and especially regarding the land uses - characteristics of the catchment and how the land uses are identified - should be improved.
The innovation in this paper is moderate and the added value should be underlined.
There are still some typos to update on the text.
